# Prioritizing areas for post-fire restoration in Greece using mixed-methods spatial analysis

Elena Palenova[1]*, Sander Veraverbeke[2,3], Igor Drobyshev[4], Themistoklis Kontos[5], Karin Ebert[1]

1 Department of Natural Sciences, Technology and Environmental Studies, Södertörn University, Stockholm, Sweden, 2 Faculty of Science, Vrije Universiteit Amsterdam, HV Amsterdam, Amsterdam, Netherlands, 3 School of Environmental Sciences, University of East Anglia, Norwich, United Kingdom, 4 Department of Southern Swedish Forest Research Centre, Swedish University of Agricultural Sciences, Alnarp, Sweden, 5 Department of Environment, University of the Aegean, Mitilini, Greece

* elena.palenova@sh.se

## Abstract

The Mediterranean region will likely face an increase in the frequency and intensity of wildfires due to climate change. Despite being fire-prone, Greece lacks a developed standardized system for identifying and prioritizing burned areas in relation to their restoration needs. Prioritization of areas for post-fire restoration efforts using Geographic Information System and Remote Sensing is a powerful decision-making tool, which, however, can be insufficient in effectively integrating socio-ecological criteria and perspectives from multiple stakeholders. Combining qualitative methods such as interviews with remotely sensed data can enhance the understanding of nuances in a local context. We designed an approach to identify high-priority areas for post-fire vegetation restoration. The identification was based on stakeholder interviews and the subsequent integration of their responses with subsequent multi-criteria overlay analysis. We selected criteria to rank the areas by interviewing 15 stakeholders working on post-fire issues. The expert interviews revealed the key characteristics respondents consider essential for prioritizing burned areas for restoration. Areas covering 77.25 km² were selected for restoration depending on the fire history, slope, and designation as part of the protected areas. Outcomes of the analysis helped to highlight three locations that potentially need special attention, with the total area of 31 km². We propose a prioritization system that is flexible, scalable, and can help government agencies, local foresters, private consultancies, and NGOs plan restoration actions and optimize the effectiveness of restoration programs in various ecosystems.

## Introduction

The understanding of current state of wildfires and their distribution worldwide is complex and skewed, as it is heavily influenced by media coverage and region-specific research efforts [1]. The total area burned globally remains relatively stable, but the

**Data availability statement:** All relevant data are within the paper and its Supporting Information files.

**Funding:** The author(s) received no specific funding for this work.

**Competing interests:** The authors have declared that no competing interests exist.

severity and frequency of extreme wildfires are on the rise due to climate change, land management and human activities [2,3]. Heatwaves and prolonged droughts create favorable conditions for large and difficult to control fires [4]. Human activities have contributed to the rising risk of extreme events, including wildfires, since the 1950s [5]. Fire plays a major role in decreasing the effectiveness of land carbon sinks already with global warming level at 1.07 °C above pre-industrial levels [6]. At 4°C, ecosystem shifts could affect 35% of the Earth's surface, increasing burned areas by 50–70% [7]. The problem is expected to escalate with a feedback loop since carbon sinks like forests will become less effective, while increased forest fires will release stored $CO_2$, thereby accelerating further climate change [7–9]. The UN Environment Programme predicts an increase in extreme wildfires of up to 14% by 2030 and 50% by 2100 [10].

Europe experienced relatively little burned area overall during recent years (2023 and 2024), but individual wildfires in countries like Greece, Spain, Italy, Portugal, France, and Scotland caused large-scale evacuations, high suppression costs, problems with water supply, damage to infrastructure and agricultural lands, and negative effects on tourism and local economies [1]. In Greece, in particular, a combination of severe fire-prone weather and abundant dry fuel heightened the likelihood of wildfires, with the probability of extreme fire seasons increasing significantly during recent years due to anthropogenic climate change [11]. The 2023 fire season was notably more severe than usual for Greece, surpassing recent fire seasons, with multiple fires burning thousands of hectares of land [12]. The country witnessed the largest wildfire in the European Union's recorded history, the Evros fire, which burned 938 km², severely affecting forests and agricultural lands and claiming 19 lives [13]. Wildfires caused extensive damage to forest and other ecosystems, infrastructure, properties, and protected areas, with significant impacts on biodiversity and local economies.

Vegetation recovery after wildfires in the Mediterranean region generally occurs without intervention, since many species are adapted to recover naturally [14,15]. Recovery rates, extent and speed are influenced by burn severity, in this case, ecological impact of fire on vegetation and soil, with higher severity leading to lower recovery rates [16]. Recovery rates are influenced by pre-fire vegetation conditions and post-fire climate, such as temperature and precipitation [17], as well as by certain topographical features, such as slope orientation: for example, vegetation on north-facing slope showed enhanced recovery rates [16].

Mediterranean countries implement emergency measures for watershed stabilization following fires every year, including the construction of check dams in streams and the installation of erosion barriers on vulnerable slopes [18]. Post-fire practices such as burned tree removal and building erosion barriers are designed to reduce soil erosion and help vegetation recover. Log erosion barriers building can reduce erosion after fire [19] and help increase diversity of recovering ecosystems [20]. Vegetation cover recovery can even slow down in some managed areas, while the natural vegetation can appear more abundant in neighboring untreated areas [20], demonstrating both the ability of local Mediterranean ecosystems to self-recover after fires and possible incorrect prioritization of areas for restoration along with the potential lack of knowledge and skills to implement the necessary actions.

Restoration in Greece after wildfires includes reforestation and soil stabilization, and is often driven by government policies, European Union funding, international collaborations, and local initiatives. Greek national public entities engaged in post-fire recovery are Ministry of Climate Crisis and Civil Protection, Ministry of Environment and Energy, Forest Service, Natural Environment and Climate Change Agency, and Ministry of Infrastructure and Transport. Post-fire management emphasizes both immediate stabilization and longer-term restoration [21]. Questions of post-fire vegetation restoration are managed mostly by the Ministry of Environment and Energy through the Forest Service, which manages and coordinates both emergency soil stabilization and long-term restoration efforts. Some other consequences such as the reconstruction of affected settlements and assets after a wildfire are managed by the Ministry of Climate Crisis and Civil Protection and the Ministry of Infrastructure [21]. Greek current policies have been criticized as being inadequate in managing post-fire soil erosion risks, indicating a need for new techniques to enhance the cost-effectiveness of land use policy interventions [18]. The National Reforestation Plan specifies certain interventions after wildfires, such as monitoring soil recovery in severely burned areas, implementation of reforestation measures, specifically, planting of saplings, upgrading of four public forest nurseries, forest and woodland clearings, and anti-erosion and flood protection actions [22]. The National Biodiversity Strategy specifies forest types that require active reforestation [23]. In addition, the National Climate Adaptation Strategy states land erosion control is a key priority area, with land stabilization measures being implemented after fires to reduce the occurrence of flooding and landslide hazards [24]. Post-fire recovery plans formally acknowledge the necessity of enhancing resilience through interventions; however, some implemented measures have been shown to either maintain or even increase wildfire risks [25].

Greece has a distinct post-fire management system, with numerous experts and involved parties offering diverse perspectives on its current state of affairs [26,27]. As wildfires impact numerous Greek stakeholders, there is a need to gather perspectives from involved parties when it comes to prioritization of areas affected by fires that need human involvement and targeted restoration efforts to help natural vegetation recovery. Stakeholder perceptions, community engagement, and the integration of ecological and social factors in wildfire management are fundamental aspects of risk mitigation and restoration activities [28–30]. The involvement of stakeholders can help in developing effective, inclusive wildfire management strategies that address both ecological resilience and community needs, enhancing the sustainability of fire-prone regions [31]. Foresters and other involved parties can guide restoration efforts by prioritizing them and help to make informed decisions about restoration actions and the allocation of financial resources. Certain stakeholders, such as government officials and policymakers, play an important role in navigating regulatory and policy challenges, including legal, zoning, and regulatory considerations that may arise during the restoration work, so their participation in discussions can enhance the effectiveness of the restoration efforts.

GIS and RS are essential components of post-disaster management prioritization frameworks worldwide [32,33]. Their ability to provide extensive spatial coverage at relatively low cost enables decision-makers to identify and prioritize affected areas without the need for extensive fieldwork. This approach offers a cost-effective and efficient method for guiding restoration strategies. In post-fire restoration contexts, remote sensing is valuable for selecting suitable restoration sites, evaluating vegetation recovery, and monitoring vegetation dynamics following restoration interventions [34]. There remains a limitation in integrating diverse models into a unified framework that can be applied at regional and national scales to support planning and decision-making processes [35]. RS addresses this limitation by providing advanced technological capabilities that facilitate the monitoring and evaluation of restoration efforts [36].

RS plays an essential role in the evaluation and monitoring of ecological restoration strategies [37]. Techniques such as time series analysis have been widely employed to assess post-fire forest recovery [38,39], including in the Mediterranean region [40,41]. RS-based prioritization of areas after fire has been applied through case studies, such as those conducted in Cyprus [42] and on Thasos island, Greece [43]. However, no studies to date have examined the entirety of Greece while also incorporating socio-ecological dimensions and insights from practitioners engaged in on-the-ground restoration projects. In organizations that are characterized by significant understaffing, aging and insufficient workforce, bureaucratic

structures, and limited financial resources employing advanced GIS and remote sensing techniques can be challenging, if not unfeasible. Nevertheless, these official services and individuals play a critical role in the decision-making processes, particularly when addressing environmental issues we are considering, such as prioritizing areas for post-fire recovery. Research involving stakeholders' perceptions of fire topics can help forest managers in their restoration efforts [31,44]. It is important to acknowledge the limitations of every method and consider the integration of complementary methods where appropriate.

Our research offers a new perspective compared to more complex traditional post-fire soil erosion risk assessments that are based on topography, vegetation, fire severity, and rainfall by integrating interdisciplinary perspectives that ensure that the study considers priorities and constraints identified by people involved in post-fire restoration. The aim of the study is to prioritize areas for post-fire restoration based on the input of involved stakeholders using a combination of qualitative and quantitative methods. Prioritizing areas for post-fire restoration in Greece according to the perspectives of practitioners may offer a novel and insightful approach and potentially reveal areas that have previously been overlooked, underexplored, or neglected.

## Materials and methods

### Methodological approach

Combining and integrating qualitative methods such as interviews with Geographic Information Systems (GIS) and Remote Ssensing (RS) can enhance the understanding of complex social and environmental issues. The combination of data types allows the exploration of nuances of different determinants of the issues, for example, the social dynamics affecting the decision-making, with mixed approaches allowing more comprehensive insights for problem analysis [45]. In this paper, we combine stakeholder interviews with GIS and RS methods to identify prioritized post-fire restoration areas in Greece.

### Interviews

We conducted 15 semi-structured interviews (Appendix 1) on topics related to stakeholders' perceptions and opinions regarding restoration management in Greece in general and their views on criteria for determining which areas should be prioritized for restoration. Discussions with the selected interviewees included post-fire restoration, such as types of interventions required following wildfires, funding allocations for restoration efforts, and strategies for implementing GIS and other technical tools. Interviews were conducted in person and lasted between 30 minutes and 3 hours, and were recorded and transcribed.

We selected interviewees in Greece from a wide range of affiliation areas, such as representatives of local and national authorities, forest agency and fire management services members, scientists, environmental non-governmental organizations (NGO) members. Stakeholders were chosen based on their expertise in post-fire environments. We conducted interviews with a total of 15 individuals. From state-level authorities, we interviewed one representative from the Ministry of Environment and Energy, who is directly responsible for coordinating restoration efforts after wildfires across Greece. At the regional level, we spoke with four representatives: two from the local Forest Service on Lesvos Island, one from the local branch of the Ministry for Climate Crisis and Civil Protection, and one from the Hellenic Fire Service. Additionally, we interviewed three researchers specializing in various aspects of forest fires, two foresters actively engaged in restoration projects, and two private sector representatives working on fire-related issues. Furthermore, we spoke with two activists from the non-governmental environmental organization The World Wide Fund for Nature (WWF) Greece, as well as one lawyer specializing in vegetation restoration and land-use changes following wildfires.

The affiliation of the interviewees is presented in Table 1 below.

**Table 1. Affiliation of the interviewees.**

| Affiliations | Number of people |
|---|---|
| State authorities | 1 |
| Local authorities | 4 |
| Forestry | 2 |
| Academia | 3 |
| Private companies | 2 |
| NGO | 2 |
| Law | 1 |

### Selecting recurrent criteria

Based on the interviews, we identified recurring criteria for prioritizing areas for post-fire recovery efforts. Only criteria mentioned by at least two different interviewees were included, as this threshold helped to show that the selected criteria reflected shared perspectives rather than isolated opinions. We counted how often each criterion was mentioned in the interview narratives (Table 2).

### Geospatial data

Based on criteria selected from the series of expert interviews (Table 2), we selected the following data for further analysis: historical data on burned areas, digital elevation model (DEM) for retrieving data on slope degrees, and boundaries of protected areas in Greece. All information used in this research was derived from publicly available data sources. Background data on the boundaries of Greece was retrieved from the Natural Earth website [46].

Historical data on burned areas for 23 years was obtained from the MCD64A1.061 Moderate Resolution Imaging Spectroradiometer (MODIS) Burned Area Monthly Global 500m data product [47], which provides spatial and temporal information on burned areas globally starting from November 2000 to the present date. It uses 500-meter spatial resolution imagery from MODIS on NASA's Terra and Aqua satellites. The product is one of the most reliable burned area datasets in Greece [48]. The data was accessed and retrieved through the Google Earth Engine (GEE) platform, which is a common tool for cloud-based processing of large-scale geospatial datasets addressing various high-impact societal issues including deforestation, natural disasters, and environmental protection [49].

The slope was derived from the Copernicus DEM product, which provides elevation data at a 30-meter spatial resolution [50]. Terrain information from this DEM dataset offers precise topographic details necessary for geospatial analysis.

The data on protected areas was sourced from the World Database on Protected Areas (WDPA), a global repository of both marine and terrestrial protected areas from the UN Environment Programme and the International Union for Conservation of Nature (IUCN) maintained by the UN Environment Programme World Conservation Monitoring Centre (UNEP-WCMC) [51]. Information on data sources, their description and usage are summarised in Table 3.

**Table 2. Selected criteria for prioritizing areas for restoration.**

| Criteria for prioritizing areas for restoration | Example of the quote from the interview | Total number of mentions of criteria in the interview text corpus |
|---|---|---|
| Repetitive fires | "Restoration should happen in double-burned areas" | 5 |
| Location in protected areas | "It is important to prioritize areas of restoration based on their protected status or archaeological or cultural significance" | 5 |
| Steep slope | "Restoration measures depend on the slope steepness" | 2 |

**Table 3. Data sources used in the geospatial analysis.**

| Data source | Description | Usage |
|---|---|---|
| MCD64A1.061, MODIS Burned Area Monthly Global 500m data product [47] | Spatiotemporal information on burned areas | For extracting historical data on burned areas |
| Copernicus DEM [50] | Digital elevation data | For extracting slope data |
| World Database on Protected Areas [51] | Information on protected areas | For extracting polygons of protected areas |
| Natural Earth database [46] | Data on administrative boundaries | For extracting background data on the boundaries of Greece |

## Computation and aggregation

### Repetitive fires

Data was processed using Python and JavaScript, and analyzed and visualized in ArcGIS Pro v3.2. First, the code was written in JavaScript for use in GEE to process the MODIS historical data on burned areas to extract and export the latest burn dates for each year from 2000 to 2024 for Greece. The dataset was filtered for each available year, the "BurnDate" band was selected, and the resulting images were clipped to the geographic extent of Greece. We selected data for 23 years, for the years 2001–2023, and the maximum burn date across the year's dataset was computed, identifying the most recent fire event for each pixel, and then final images were exported as a GeoTIFF file (See code in Appendix 2).

Then, repetitive burned area occurrence throughout the years was calculated using Python. Raster data related to burned areas in Greece from MODIS over a period from 2001 to 2023 were analyzed as overlay analysis with NumPy library for raster calculation, summing identified burned areas on multiple raster files (Appendix 3). For further analysis, repetitive fires were reclassified using the "Reclassify" tool in ArcGIS Pro.

### Slope

Slope is frequently mentioned as the most important topographic factor for landslide susceptibility in Greece [52–54], with landslide frequency and density increasing significantly on slopes exceeding 15° [55,56]. Landslide prevalence was found to be linked to specific slope ranges in the Mediterranean [57], suggesting that lower-bound thresholds are important for identifying areas susceptible to issues related to vegetation restoration after wildfires. Slope data for Greece was derived from DEM using the "Slope" tool in the "Spatial Analyst Tools" in ArcGIS Pro. Areas with slopes steeper than 15° were reclassified using the "Reclassify" tool.

### Protected areas

Protected area polygon boundaries were reclassified into raster format using the "Feature to Raster" tool from the "Conversion Tools" in ArcGIS Pro.

### Overlay analysis

Using selected and reclassified datasets, we performed overlay analysis using Raster Calculator spatial analyst tool with "Maximum of Inputs" cell size to identify areas that could be prioritized for restoration. Since input rasters had different spatial resolutions (30 m and 500 m), we first resampled the 30 m raster to 500 m using the spatial averaging method. The final output represents a composite suitability map at 500 m resolution. The analysis involved preselecting areas that met at least two out of three criteria (Table 2), followed by the identification of final priority areas where all three datasets overlapped. We employed the "Zonal Statistics" tool to quantify the number of pixels within each area.

## Ethics statement

The study used verbal interviews to gather data and documented them using audio recordings. Recruitment period for this study was carried between June 6 and August 11, 2024. Our research did not include minors. We received the approval from Södertörn University ethics committee to carry out the study with the recommendation from the University Council of not providing additional ethical screening. Therefore, no ethical concerns were identified in the research process. All methods adhered to standard scientific practices and guidelines. All participants involved in this study gave informed consent to participate in the study. Participants were informed that participation was voluntary, and they could withdraw from the study at any time without penalty. All responses were kept confidential, anonymized, and stored securely, accessible only to the research team. Conclusion of the ethics committee is included as additional file to the manuscript submission.

## Inclusivity in global research

Additional information regarding the ethical, cultural, and scientific considerations specific to inclusivity in global research is included in the Supporting Information (S2 File).

## Results

We calculated the frequency of wildfire events in Greece from 2001 to 2023 (Appendix 3) and identified 15207 km² of areas that were burned at least once during this period (Fig 1).

Then we computed (Appendix 4) 2598 km² of repetitively burned territories for 23 years (Fig 2), therefore, out of the total burned area, 17% of the area was burned repetitively, twice or more.

After retrieving the elevation data and classification (Appendix 5) and reclassification of the slope, areas were selected where the slope exceeded 15° (Fig 3). 42169 km² were selected which is around 32% of the area of Greece.

Based on data provided by UNEP and IUCN, Greece has 1289 total protected areas, both terrestrial and marine, with 46224 km² of land area covered, meaning that 35% of the country is terrestrial and inland waters protected area coverage [51]. They cover most of the IUCN management categories, however, 37% of areas are not reported; most protected areas are federal or national ministry or agency-governed. The distribution of protected areas is shown in Fig 4.

We identified preliminary areas with any of rasters overlapping, with final selection as the result of overlap analysis of all data of areas prioritized for restoration (Fig 5).

Finally, 77.25 km² were selected at the end for prioritization in total, with the following distribution by regions (Table 4).

Of the 14 regions in Greece, including the autonomous region of Mount Athos, 12 were selected as having areas prioritized for restoration, with no areas selected in West Macedonia or Mount Athos.

## Discussion

### Selected areas

Based on all the areas identified in the results section, we would like to emphasize three specific territories (Fig 4) with selection based on distinct criteria, that deserve special attention for targeted restoration efforts following wildfires. The first is the region on the southern part of Evia (or Euboea), chosen due to its dense and flammable forest and exposure to the dry strong summer winds which makes it susceptible to recurrent fires and extensive burned area. The second is the Evros region, a frequently fire-affected area that attracts significant media attention due to its border location with Turkey and its recurring intense fire incidents. Lastly, we highlight the area around Athens, selected due to its proximity to the capital and the large population density.

 

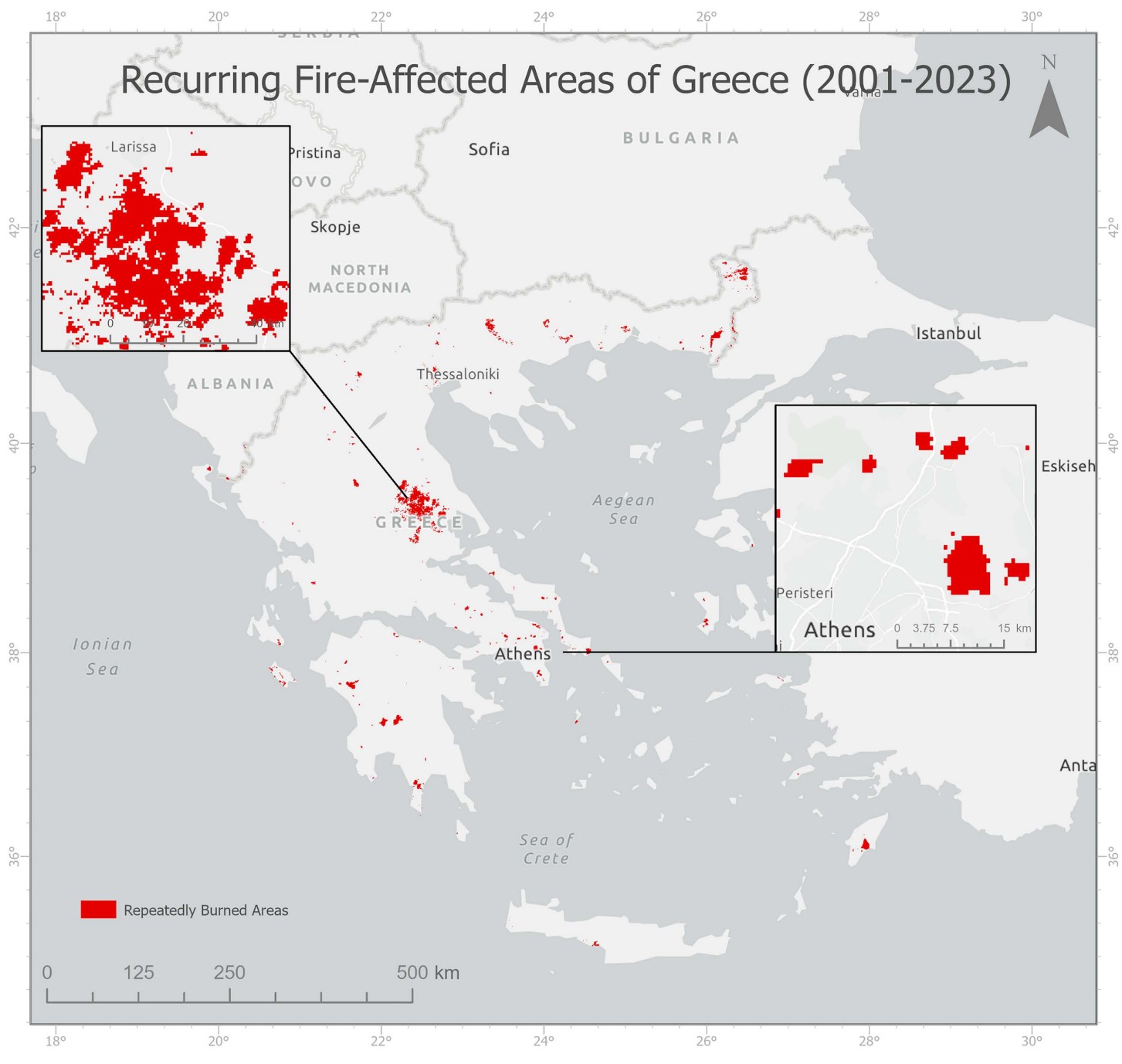

**Fig 1. Wildfire Frequency in Greece (2001–2023).**

## Evia

Wildfires are a recurrent phenomenon on Evia island, located in Central Greece region. A particularly devastating wild-fire occurred in August 2021 was described as "catastrophic" and "unprecedented in recorded history" [58]. This event resulted in the burning of more than 2000 km² of area, leading to severe consequences including the complete destruction

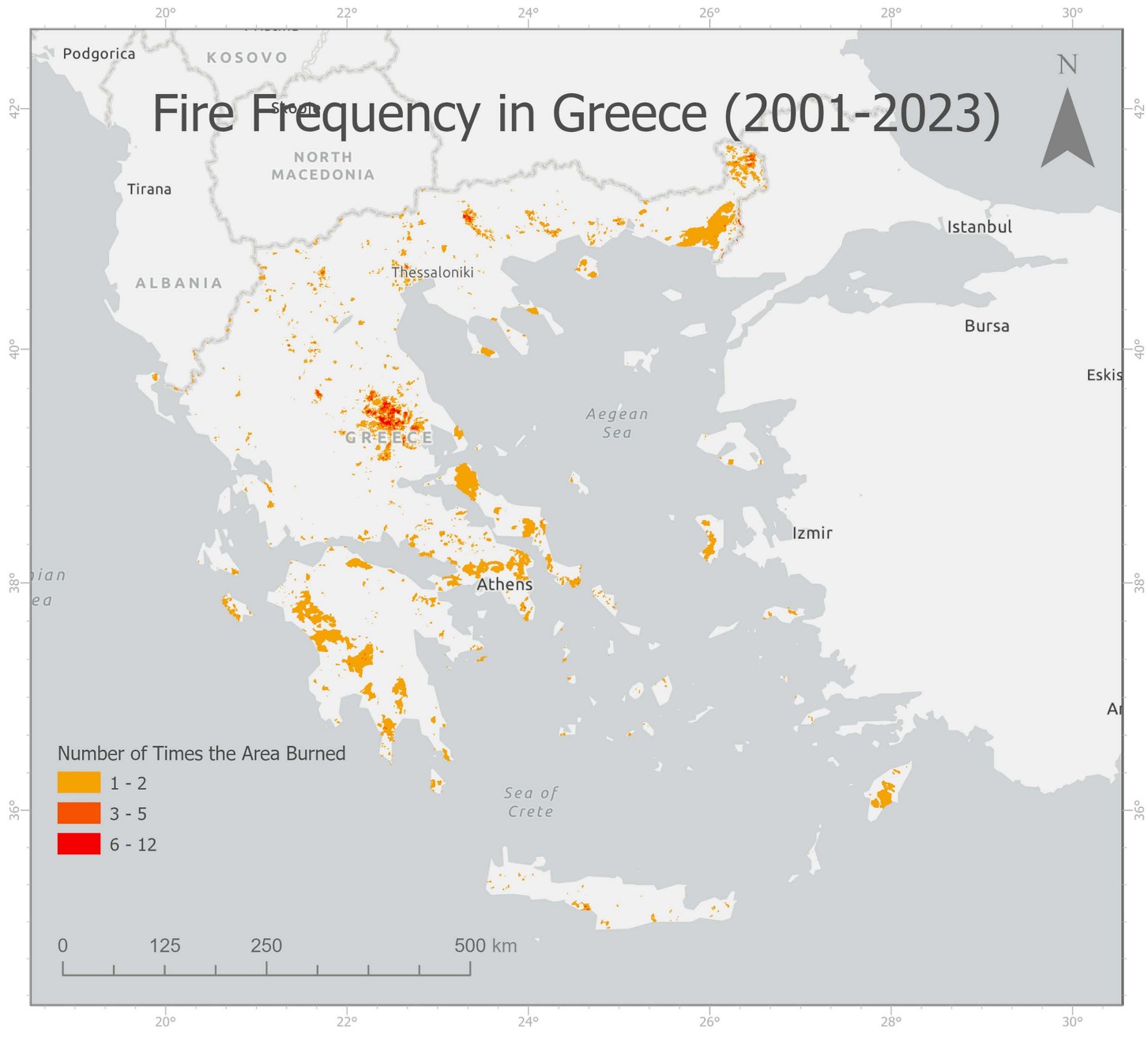

**Fig 2. Repeatedly Wildfire-Affected Areas of Greece (2001–2023).**

of several small villages and agricultural areas. The flammability of this region, besides the reasons typical for all of Greece such as the fire-prone Mediterranean climate and vegetation, can be explained by the Evia's dense forest cover, predominantly composed of pine trees, which are highly flammable due to their resin content, and to exposure to the strong summer winds that can rapidly spread fires across large areas.

Selected area for prioritization covers 18.25 km² and is located at the "Mountain Ochi, coastal zone and islands" protected area. It is designated on a regional level through the Birds Directive in 2010 with 334 km² or reported protected area. Post-fire

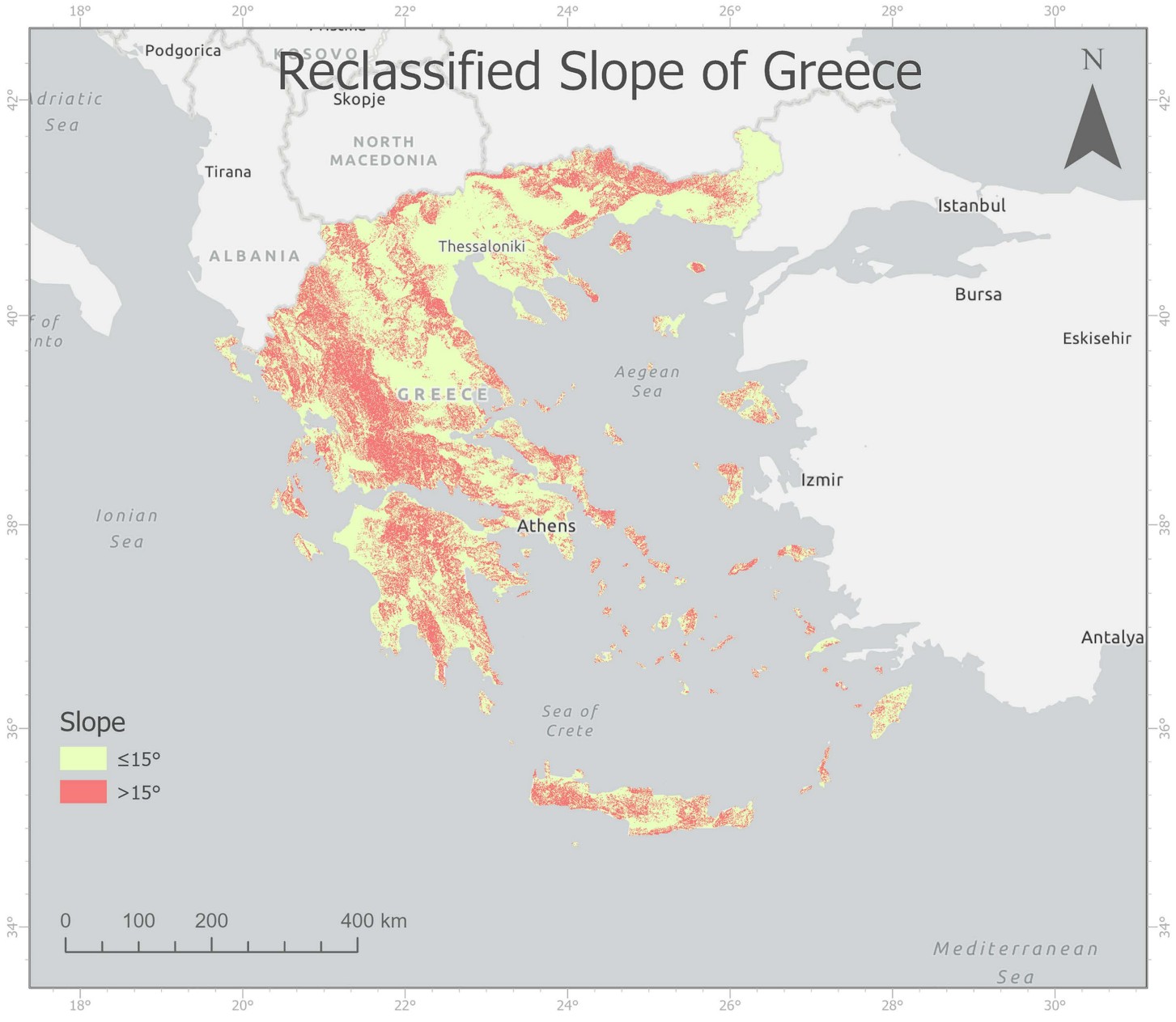

**Fig 3. Reclassified slope of Greece.**

management in Ochi mountain range is important due to its rich biodiversity and rare endemic species: 57 species are protected in the area [59]. The area's steep terrain and dense vegetation can increase the risk of soil erosion and flooding if not properly restored after fires. Effective restoration in Ochi can help preserving both its environmental value and local livelihoods.

## Evros

Next area selected for prioritization where repetitive fires happen on protected territories on steep slopes covers 9.25 km² and is located in East Macedonia and Thrace region in northeastern Greece. The areas affected by the recurring fires in the Evros regional unit there among others fall under the Dadia-Lefkimi-Soufli Forest national park designation. The

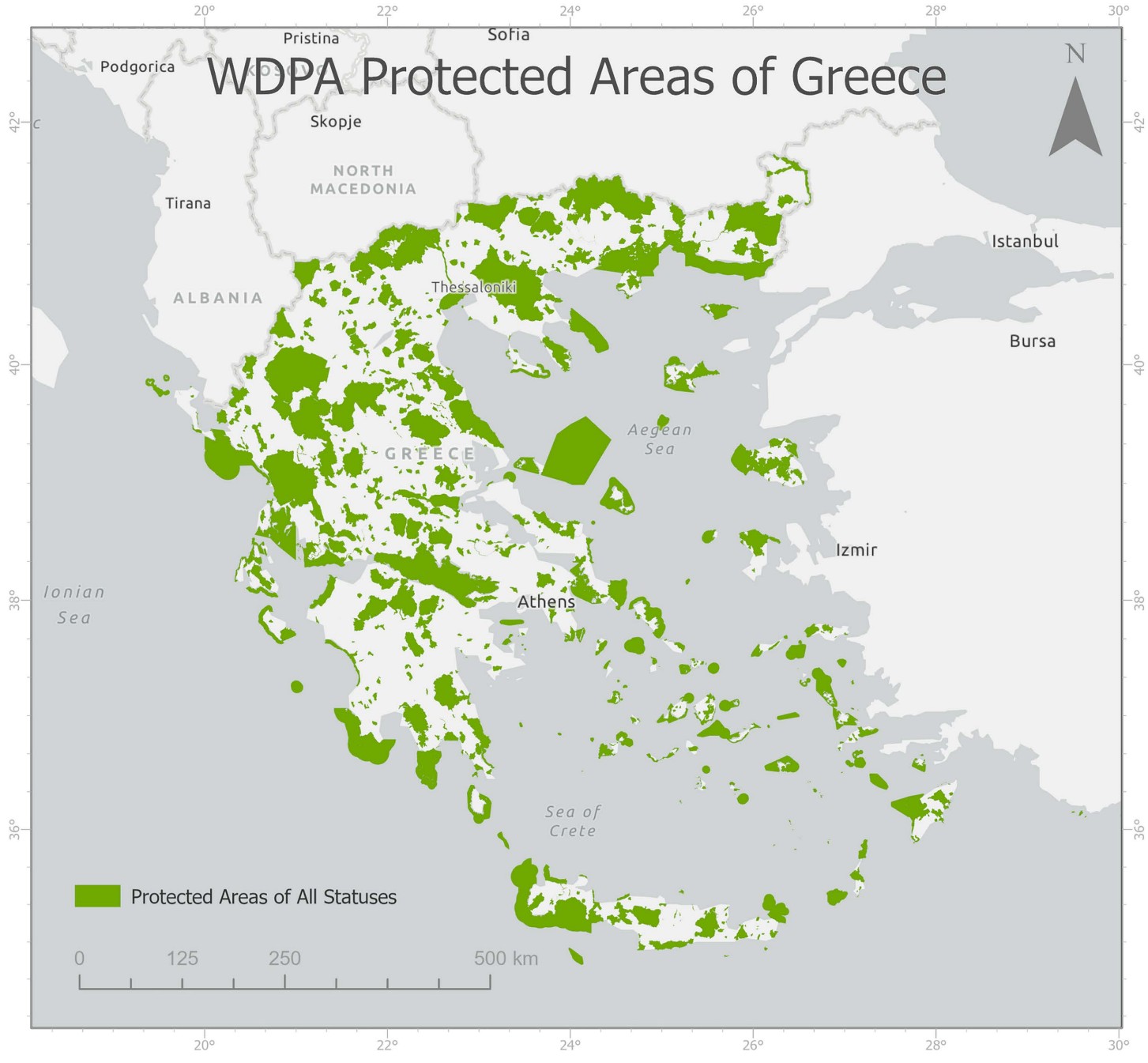

**Fig 4.  Greek protected areas according to the World Database of Protected Areas (WDPA).**

reported area of the national park is 432 km² and the IUCN management category is VI "managed resource protected areas", meaning that there is a possibility of sustainable extraction of natural resources with activities allowed such as forestry, fishing, or agriculture [60].

In 2023 Greece experienced an extremely severe fire season, with the total burned area exceeding the country's historical averages and the Evros fire being the largest wildfire in the European Union's recorded history [13]. 938 km² were burned, 19 people died, forests, agricultural lands, infrastructure, and properties were extensively damaged.

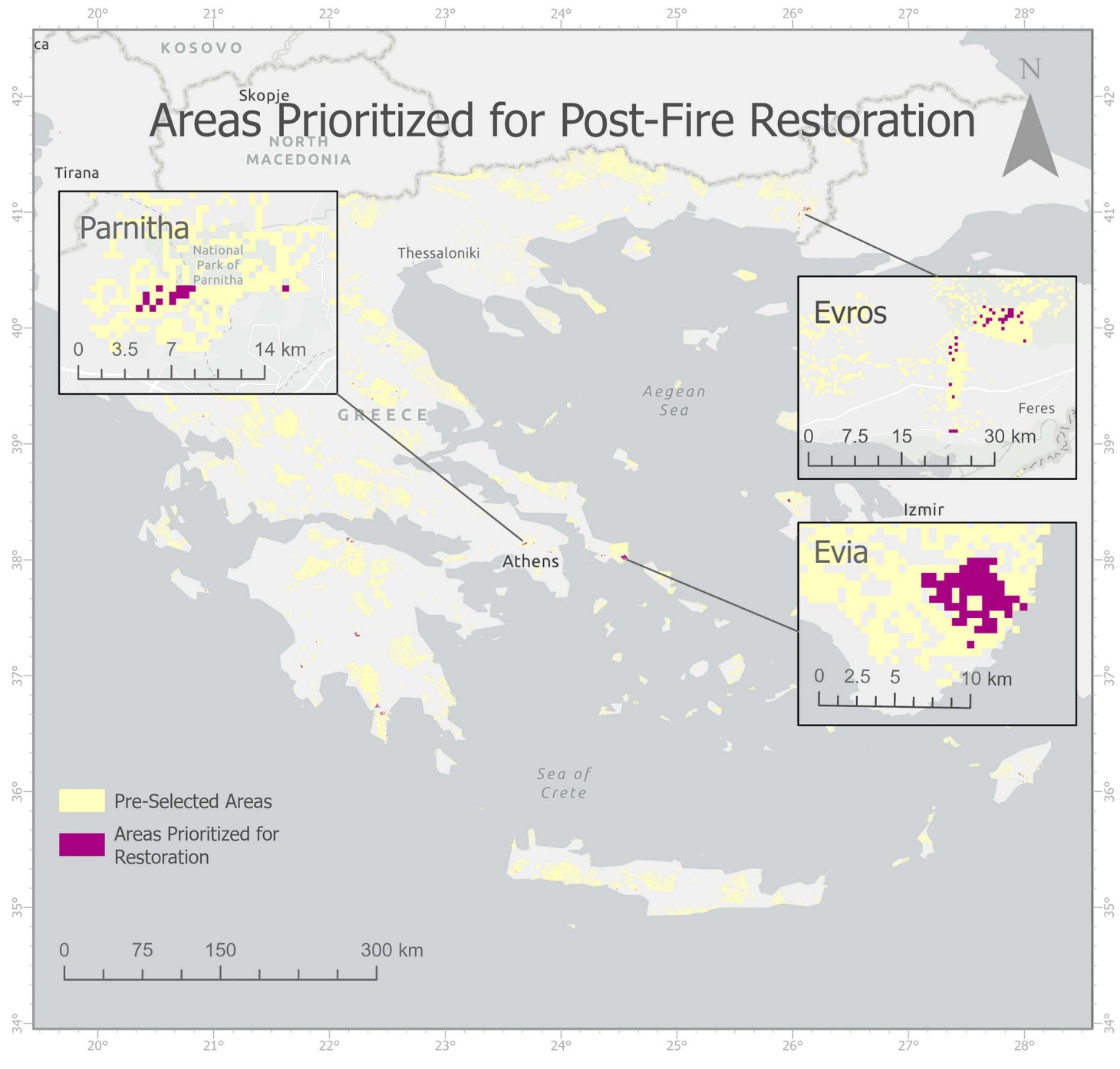

**Fig 5. Selected Areas Prioritized for Post-Fire Restoration.**

The Evros region experiences extensive repetitive fires in the area of the Dadia-Lefkimi-Soufli Forest national park that is covered by pine trees with highly flammable foliage [61]. The density of forest vegetation can also affect limitations of accessibility in firefighting efforts, and fires used in agricultural practices for land clearing can spread uncontrollably and trigger wildfires. Evros' border location with Turkey can complicate the coordination for firefighting. Land-use practices can

**Table 4. Size of areas selected for prioritization with distribution by regions, in km².**

| NUMBER | REGION | AREA SIZE (km²) |
|---|---|---|
| 1 | Peloponnese | 22.75 |
| 2 | Central Greece | 18.5 |
| 3 | East Macedonia and Thrace | 9.25 |
| 4 | West Greece | 7 |
| 5 | North Aegean | 5.25 |
| 6 | Attica | 4.75 |
| 7 | South Aegean | 2.75 |
| 8 | Thessaly | 2.25 |
| 9 | Epirus | 1.5 |
| 10 | Crete | 1.5 |
| 11 | Ionian Islands | 1.25 |
| 12 | Central Macedonia | 0.5 |
| | **TOTAL** | **77.25** |

contribute to fire risk, and also, as a border area, it is a place for military activity and possible accidental fires caused by training exercises, equipment, or munitions.

Wildfires in Evros were studied as both natural and political phenomena, emphasizing the media's critical role in shaping public perceptions and responses to such events [62]. Greek media framing of the wildfires was found to be influenced by national populism and xenophobic attitudes, since narratives of blame were shifted to, among other reasons, migrant activities, arson, and war. Evros area is notable particularly in the context of the European refugee crisis and areas affected by fires therefore need special attention when it comes to post-fire management.

## Parnitha

Areas that were selected for restoration on the southeast side of the Greek mainland, in the Attica region, among other places, are located close to Athens on the territory of the National Park of Parnitha and cover an area of 3.5 km². Parnitha National Park is 39.5 km² large area and has IUCN management category II status. Generally speaking, IUCN Category II protected areas are national parks where activities like industrial development, extensive agriculture, and resource extraction are prohibited, but where it is possible to have scientific research and environmentally friendly tourism [60].

Along with other factors that are relevant to the entire territory of Greece, Parnitha can experience exacerbated fire activity due to the high human presence in the area, leading to accidental ignition from activities such as camping, picnicking, or cigarette butts. Past fires, such as the one in 2007, have disrupted the forest's natural recovery process and affected the forest's genetic diversity [63], with dry dead vegetation creating additional fuel that can increase the likelihood of repetitive fires. The recovery processes and the effectiveness of reforestation efforts in Parnitha have been the focus of extensive research, potentially attributed to its close proximity to the Greek capital and the suitability of the territory for post-fire research, making scientific work there easily accessible and demonstrative [64–67].

## Restoration measures

Among post-fire actions are those minimizing environmental and social impacts, helping the ecosystems to restore, and preventing future fires [68]. The most important action should be assisting natural regeneration with appropriate consequential fuel management. It can be done by limiting extensive agricultural activities and land use in burned areas to prevent further damage to recovering vegetation and soil. For immediate after-fire response, soil stabilization measures can be implemented, by using erosion barriers to reduce runoff and soil loss on slopes [68]. The area might be cleared of the

dead trees and brush after fires to reduce fuel for future fires to prevent repetitive fires. Replanting can be implemented in certain areas specifically pre-selected for it since this measure can be time- and cost-consuming. Community engagement and education should be used to deal with the consequences of repetitive wildfires in vulnerable areas, with involvement of locals in reforestation and soil conservation efforts, funds allocation for participating in restoration efforts, and for education in post-fire recovery techniques [68].

It is crucial that any restoration measures, particularly those that are costly and extensive, be undertaken in collaboration with scientists and local forestry practitioners. Their expertise is very important in determining the most appropriate actions for a given site. In some cases, such as areas affected by fire, it may be unnecessary to allocate resources if the optimal action plan is simply to allow the area to recover naturally without any intervention, but protecting areas from land use change and anthropogenic economic activity, such as agriculture, grazing or construction building.

## Additional insights from the interviews

Eight participants of the interview regarding post-fire restoration measures highlighted the issue of understaffing within the Forest Service. Respondents raised these concerns as "We need more workers to protect affected areas", "Forest Service has lack of personnel", "We should hire educated people", "The evaluation of burned areas is suffering from the lack of personnel in Forest Service". Interviewees expressed the necessity of hiring young educated personnel, six persons noted that saying about the "necessity to involve young people", the "need to hire more young people who know technologies in the Forest Service". Also, the Forest Service representative mentioned the necessity of "transfer of knowledge from old experienced staff to new staff to prevent the knowledge gap". It is important to note that in the Forest Service headquarters in Athens, only two individuals are responsible for all-Greek coordination and management of fire restoration efforts across the entire country, making the shortage of personnel a critical challenge in the context of post-fire restoration.

Four respondents also mentioned the use of technologies such as GIS as essential for post-fire restoration efforts; however, they noted that these technologies are either not used or not sufficiently implemented in practice. "Staff of services needs to know spatial analysis, GIS, photogrammetry", "We need to supervise the area with new technologies, drones, GIS", mentioned interviewees. One respondent mentioned that "Young people need to be involved outside in the field and inside offices, with knowledge of GIS because", another respondent said that "It is harder to implement new technologies like GIS in aging services, hard to teach old dogs new tricks".

Issues related to funding were highlighted by seven respondents, who specifically pointed out the misallocation of resources. Concerns included underfunding of the Forest Service's efforts, excessive funding allocated to the Fire Service responsible for fire suppression, and a disproportionate focus on fire suppression equipment rather than on training and hiring personnel to implement preventive and post-fire measures critical for effective fire management. "The main problem of restoration is underfunding", "The way of money allocation is irrational, the government prefers to buy a new fire truck, but not to spend money for people's education", "There is an underfunding in the Forest Service, especially when it comes to restoration, overfunding in the Fire Service", mentioned some of the respondents.

Five respondents also emphasized the absence of a holistic fire management approach, which they argued is essential for effectively addressing fire-related challenges in Greece. This was attributed to a lack of communication between the agencies responsible for fire management and the absence of a unified action plan, including a systematic framework for identifying vulnerable areas that needed attention after fires and implementing targeted restoration measures, while nowadays the restoration actions are sporadic and random. "There is a lack of holistic approach in wildfire management in Greece", "Consequences of wildfires is not a one-dimensional problem, it is complex and needs a holistic approach", "Ministries are not doing restoration in an organized way, there is no methodology implemented", the interviewees mentioned among other things. One respondent highlighted the need to create and adapt the methodology to categorize and prioritize burned areas that need attention, which is the aim of this research.

## Methodological remarks

Recent studies confirm both the necessity of the chosen methodology and the prioritization criteria obtained as a result of the analysis. Multi-criteria decision-making methods are increasingly applied to systematically weigh and aggregate criteria [69]. Studies have shown that the selection of criteria for post-fire restoration prioritization incorporates a mix of biophysical, ecological, and socio-economic factors. Common criteria found in the literature include fire frequency, topographical features such as slope, land cover types, proximity to human settlements, presence of protected areas, and vegetation characteristics [70,71]. Therefore, our selection for the prioritization of repetitive fires, slope, and protected areas thus closely aligns with current trends and recommendations in the research literature.

The mixed-method approach offered in our study provides a different perspective compared to an analysis based only on a purely quantitative approach, with the calculation based on topography, vegetation cover, fire severity, and rainfall. A more elaborate approach would likely provide a more precise assessment of post-fire restoration areas, however, its practical application may be constricted by the institutional capacity. Our focus is on double-burned areas on steep slopes and protected areas, and this methodological choice is relevant in a context where decision-making is limited by understaffing, bureaucracy, and lack of financial resources. Combining qualitative data with geospatial analysis offers a more nuanced understanding of the socio-environmental dynamics influencing post-fire restoration priorities. Our approach ensures that restoration efforts align with practical constraints and local context.

The proposed methodological approach can be an initial step in identifying priority areas for post-fire restoration through geospatial analysis informed by interviews. While factors such as fire severity, timing between fire events, and broader ecological characteristics were not considered due to the fact that they were not mentioned during interviews with practitioners, enhancing spatial resolution and incorporating additional variables like aspect, soil moisture, vegetation type, and land use can be a starting point for more comprehensive research. This methodology could be further improved by incorporating higher-resolution burned area data from Landsat or Sentinel (10–30 m) instead of 500 m to enhance accuracy. Additionally, integrating other variables recognized as significant in the research literature, such as aspect, fire severity spectral indices, e.g., differenced Normalized Burn Ratio (dNBR), and vegetation type/land cover, would further improve the analysis. Although these variables were not identified through the interviews, existing research literature shows their importance in post-fire ecological assessments.

The scalability of the offered methods can also be considered as a potential strength. While applied for Mediterranean region, methods consider the natural regeneration potential of the region and on-the-ground socio-ecological limitations, with natural recovery of vegetation after wildfires often does not require additional human efforts due to the occurrence of the fire-adapted species. Despite its simplicity and limited sample size, this research offers a clear and focused approach which can be also scalable and suitable for application across various areas. Interviewing stakeholders working in different types of ecosystems and environmental conditions can introduce new criteria for prioritizing areas for vegetation restoration, and these insights may influence subsequent geospatial analyses and findings.

## Framework validation and data gaps

The potential for future work lies in validating the proposed framework by comparing the identified priority areas with the outcomes of existing restoration actions in Greece. Currently, comprehensive data on such areas are absent and not provided by official authorities, as confirmed through document reviews, online research, and an interview with a representative from the Greek Forest Service. No official openly available maps exist to indicate the exact location or extent of these areas. Therefore, the validation and mapping of restored areas remain challenging. Advancing research efforts to collect, systematize, and openly share this information, and to apply it in validating the framework presented in this study, would significantly strengthen its practical applicability and methodological robustness.

## Conclusions

We show that a prioritization approach based on mixed-methods spatial analysis can be used to define and visualize the potential targets of large-scale restoration programs. Visualization based on our analysis can be used for developing targeted recommendations for selected areas requiring restoration. This prioritization method is flexible and can help efficiently in different ecological and socioeconomic contexts. Using a combination of qualitative and quantitative methods helps to consider the natural regeneration potential of Mediterranean ecosystems and local socio-ecological limitations, and can help government authorities, foresters, private companies, and environmental NGOs plan restoration actions and optimize the effectiveness of restoration programs. It can be used as a supporting decision-making tool for restoration planning in various regions.

## Supporting information

**S1 Fig. Fire Frequency in Greece (2001–2023).**
(TIF)

**S2 Fig. Repeatedly Fire-Affected Areas of Greece (2001–2023).**
(TIF)

**S3 Fig. Reclassified slope of Greece.**
(TIF)

**S4 Fig. Greek protected areas according to the World Database of Protected Areas (WDPA).**
(TIF)

**S5 Fig. Selected Areas Prioritized for Post-Fire Restoration.**
(TIF)

**S6 Fig. Appendix 5. Slope of Greece.**
(TIF)

**S1 File. Supporting information.** Appendices 1–5.
(DOCX)

**S2 File. Inclusivity in global research questionnaire.**
(DOCX)

**S3 File. Minimal Data Set.**
(ZIP)

## Acknowledgments

We thank the stakeholders involved in this study for their willingness to share their knowledge and time and for their engaging motivation.

## Author contributions

**Conceptualization:** Elena Palenova.

**Data curation:** Elena Palenova, Themistoklis Kontos.

**Formal analysis:** Elena Palenova.

**Investigation:** Elena Palenova.

**Methodology:** Elena Palenova.

**Project administration:** Elena Palenova.

**Resources:** Elena Palenova.

**Supervision:** Sander Veraverbeke, Themistoklis Kontos, Karin Ebert.

**Validation:** Elena Palenova.

**Visualization:** Elena Palenova.

**Writing – original draft:** Elena Palenova.

**Writing – review & editing:** Elena Palenova, Sander Veraverbeke, Igor Drobyshev, Karin Ebert.

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
