## [Editor Report · Decision Letter 0]

17 Jun 2025

Dear Dr. Palenova,

Please include continuous line numbers in your submission, as these are essential to ensure a smooth review by the reviewers.

Please also include figures within the text in Word format (you may include these figures at the end of the text).

This decision is not based on the scientific quality of the manuscript, but solely on technical limitations that prevent peer review. 

We look forward to receiving your revised manuscript.

Kind regards,

Alberto Alaniz

Academic Editor

PLOS ONE

Journal Requirements:

4. Please provide additional details regarding participant consent. In the ethics statement in the Methods and online submission information, please ensure that you have specified what type you obtained (for instance, written or verbal, and if verbal, how it was documented and witnessed). If your study included minors, state whether you obtained consent from parents or guardians. If the need for consent was waived by the ethics committee, please include this information.

5. Please update your submission to use the PLOS LaTeX template. The template and more information on our requirements for LaTeX submissions can be found at http://journals.plos.org/plosone/s/latex.

6. We note that Figure 1, 2, 3, 4, 5 and Appendix 5 in your submission contain [map/satellite] images which may be copyrighted. All PLOS content is published under the Creative Commons Attribution License (CC BY 4.0), which means that the manuscript, images, and Supporting Information files will be freely available online, and any third party is permitted to access, download, copy, distribute, and use these materials in any way, even commercially, with proper attribution. For these reasons, we cannot publish previously copyrighted maps or satellite images created using proprietary data, such as Google software (Google Maps, Street View, and Earth). For more information, see our copyright guidelines: http://journals.plos.org/plosone/s/licenses-and-copyright.

a. You may seek permission from the original copyright holder of Figure 1, 2, 3, 4, 5 and Appendix 5to publish the content specifically under the CC BY 4.0 license.

7. Please remove all personal information, ensure that the data shared are in accordance with participant consent, and re-upload a fully anonymized data set.

Additional Editor Comments 

Dear Elena Palenova

I hope you are well. I have reviewed the manuscript “Prioritization of areas for restoration after fires in Greece using spatial analysis with mixed methods,” which has several formatting issues that prevent it from being sent for peer review. Ths desicion is due solely to formatting restrictions, because the manuscript has not been yet peer reviewed (https://journals.plos.org/plosone/s/submission-guidelines).

Please include continuous line numbers in your submission, as these are essential to ensure a smooth review by the reviewers.

Please also include figures within the text in Word format (you may include these figures at the end of the text).

This decision is not based on the scientific quality of the manuscript, but solely on technical limitations that prevent peer review. Best regards

---

## [Author Response · Author response to Decision Letter 1]

17 Jul 2025

Response to the Academic Editor:

Dear Ronalyn,

I was given 21 days for revisions and 10 days passed without you answering my previous questions. I do not know what to do because I came a cross a bottleneck for the article revision.

My manuscript information is PONE-D-25-26522R1 [EMID:85b8de9c8da75e44]

Name: Prioritizing Areas for Post-Fire Restoration in Greece Using Mixed-Methods Spatial Analysis

In the current revision, there is a concern regarding 30 m Copernicus Digital Elevation Model (DEM). As I mentioned in my letters via submission form and via emails, the 30m DEM from Copernicus license is Copernicus Free and Open Data License.

I also contacted Copernicus Data Space Ecosystem Support Team specifically regarding signing up the form. They said the following:

"Unfortunately, the procedure asked (signing the form) is not foreseen. Please contact PLOS One journal and state that "At CDSE, there's no known procedure for signing such a document. A user that accepts ESA User License, is eligible to use CCM data according to the terms and conditions written in the ESA User License. I have accepted the ESA User License, you can find it here: https://dataspace.copernicus.eu/sites/default/files/media/files/2025-06/copernicus_contributing_mission_data_access_v2_cop_dem_licenses.pdf hence I'm eligible use this data for scientific purposes. Accordingly no further acknowledgment in the forms signature from CDSE or ESA or CCM personnel is possible or being provided".

Kind regards, Copernicus Data Space Ecosystem Support Team"

If you need screenshots of my correspondence with Copernicus, I can provide them.

Please let me know what I should do.

Best regards,

Elena Palenova

Previous response to Reviewers:

PLOS ONE

PONE-D-25-26522

Prioritizing Areas for Post-Fire Restoration in Greece Using Mixed-Methods Spatial Analysis

Response to Reviewers

by Elena Palenova

Regarding previous comments by Ronalyn M. Ramos:

1. I uploaded data received through the GIS analysis as the Supporting Information file, zip-file.

2. I removed figures from my manuscript. You have already asked me to remove it during the first editing, then asked to put back, and now asked to remove again. I will do anything according to guidelines, just let me know.

3. All datasets used in figures are publicly available and can be used in research with proper referencing, which meets PLOS ONE requirements for CC BY 4.0 license. No copyrighted base maps or proprietary data (e.g., Google Maps, Copernicus satellite imagery under restrictive terms) were used.

- MCD64A1.061 MODIS Burned Area Monthly Global 500 m by NASA LP DAAC; license is public domain, use is unrestricted, including for redistribution and publication.

- Copernicus Digital Elevation Model (DEM) by DLR & Airbus (2010–2018); license is Copernicus Free and Open Data License, use is free for commercial and non-commercial purposes with proper citation.

- World Database on Protected Areas (WDPA) by UNEP-WCMC; license is free to use for non-commercial purposes (including academic research), use is permitted with proper attribution, with no prior permission required for academic use.

- Natural Earth from naturalearthdata.com; license is public domain; use is free for any purpose—personal, academic, or commercial—with no restrictions on modification, redistribution, or publication.

Attribution is provided within the manuscript, please let me know if extra references on the figures, on the maps themselves should be removed! I will do it.

Regarding Additional Editor Comments:

1. I included continuous line numbers in my submission

2. I included figures within the text in Word format

Regarding Journal Requirements:

1. I changed the text formatting based on body formatting guidelines and title, author, affiliation formatting guidelines, added supporting information and changed names of attached files. Let me know if I am missing something and I will make all changes!

2. I included the questionnaire on inclusivity in global research as supporting file and added information to the Methods section.

3. Code sharing information is given in Declarations - Data Availability Statement. In “Recommendations for sharing code” it says it is “encouraged” to share the code in repository with DOI and licenses etc., but not obliged, so I would prefer to have a code as it is now, in Appendices, and in Data Availability Statement of the manuscript mentioning how the code can be accessed. If it is an obligation, please let me know and I will follow all necessary directions!

4. Details regarding paritcipant consent were added both to Methods section and “Ethics Statement” field of the submission form in the previous review.

5. The submission follows the LaTeX format, however, it is a bit confusing regarding changing the Word format to PDF and recommendations such as "do not include figures in your PDF; they should be uploaded as separate files", because they contradict instructions of submitting Word file with tracked changed and include Figures there. Also, there are some differences in LaTeX format and Manuscript body formatting guidelines I followed, for example, Supporting information being before or after References and many other small differences. Maybe I am mixing up things, but please let me know which directions to follow, and I will do everything that is needed!

6. Data used in Figures 1-5 and Appendix 5 are from open sources/in public domain that can be used under the Creative Commons Attribution License (CC BY 4.0). It includes:

- Digital elevation model (DEM) from Copernicus

- MCD64A1.061 MODIS Burned Area Monthly Global 500m data product from NASA

- Boundaries of protected areas from the World Database on Protected Areas

- Administrative boundaries from the Natural Earth database

All used data is available under CC BY 4.0 license and properly cited in the manuscript. Maps were created using ArcGIS Pro software licensed through my university. Google Earth Engine was used only as a tool of retrieving MODIS (NASA) data, but not for creating figures for the article. Let me know if I need to edit maps, remove background copyrights available there from the basemap, etc.

7. All personal information is removed already, submitted file is anonymized.

8. I included captions for my Supporting Information files at the end of my manuscript. In the text body, figures are named as "Fig 1. Fire Frequency in Greece (2001-2023).", in Supporting Information at the end, they are named "S1 Fig. Fire Frequency in Greece (2001-2023)." I hope it is correct way to do it, but let me know if no, and I will edit and fix everything!

Regarding other comments:

I uploaded files to https://pacev2.apexcovantage.com/ and replaced original source files with PACE generated files, both in Word and as attached supporting files.

---

## [Decision Letter · Decision Letter 1]

26 Nov 2025

Dear Dr. Palenova,

Thank you for submitting your manuscript to PLOS ONE. After careful consideration, we feel that it has merit but does not fully meet PLOS ONE’s publication criteria as it currently stands. Therefore, we invite you to submit a revised version of the manuscript that addresses the points raised during the review process.

**ACADEMIC EDITOR:**

We look forward to receiving your revised manuscript.

Kind regards,

Kristofer Lasko, PhD

Academic Editor

PLOS ONE

**Note from Editorial Office:**

Having contacted Reviewer #2, we have confirmed that their use of AI was for translation purposes only, which is within the PLOS One policy on AI use. Hence, we require that you respond in full to the comments of Reviewer #2.

**Journal Requirements:**

Reviewers' comments:

Reviewer's Responses to Questions

**Comments to the Author**

Reviewer #1: (No Response)

Reviewer #2: All comments have been addressed

2. Is the manuscript technically sound, and do the data support the conclusions?

Reviewer #1: Yes

Reviewer #2: Partly

3. Has the statistical analysis been performed appropriately and rigorously?

Reviewer #1: Yes

Reviewer #2: N/A

4. Have the authors made all data underlying the findings in their manuscript fully available?

Reviewer #1: Yes

Reviewer #2: Yes

5. Is the manuscript presented in an intelligible fashion and written in standard English?

Reviewer #1: Yes

Reviewer #2: Yes

Reviewer #1: This manuscript addresses a highly relevant and timely topic, namely the prioritization of areas for post-fire restoration in Greece using a mixed-methods approach that integrates stakeholder perspectives with GIS and remote sensing analysis. The study is well-structured, uses openly available datasets, and offers practical insights that could inform restoration planning and policy. Its novelty lies in combining expert interviews with geospatial analysis, which provides a socio-ecological dimension often missing from purely technical frameworks.

However, the manuscript would benefit from further strengthening in the Introduction, Methods, and Discussion to better situate the work in the existing literature and to clarify its methodological contributions.

The Introduction section needs to be enhanced. At present, it provides a good overview of wildfire impacts and post-fire management in Greece but lacks sufficient contextualization of the role of GIS and remote sensing in prioritization frameworks. I recommend adding a dedicated paragraph on the use of GIS-based approaches for restoration prioritization, referencing relevant international and Mediterranean studies. This will help position your mixed-methods approach in relation to existing geospatial methodologies and highlight the novelty of integrating stakeholder input.

Some suggested references you could include are:

• Dosis, Stefanos, George P. Petropoulos, and Kleomenis Kalogeropoulos. "A geospatial approach to identify and evaluate ecological restoration sites in post-fire landscapes." Land 12.12 (2023): 2183.

• Prodromou, M.; Gitas, I.; Mettas, C.; Tzouvaras, M.; Themistocleous, K.; Konstantinidis, A.; Pamboris, A.; Hadjimitsis, D. Remote-Sensing-Based Prioritization of Post-Fire Restoration Actions in Mediterranean Ecosystems: A Case Study in Cyprus. Remote Sens. 2025, 17, 1269. https://doi.org/10.3390/rs17071269

The current subtitle “Mixed methods” could be revised to a more professional and descriptive term. I recommend using a title such as “Methodological Framework”, “Study Design and Analytical Methods”, or “Methodological Approach”. This will better reflect the integration of qualitative (stakeholder interviews) and quantitative (GIS/RS analysis) components and improve the overall readability of the manuscript.

Discussion – Methodological Remarks

In lines 513–525, where you discuss the methodological remarks, I recommend strengthening the comparison with other studies that have addressed the selection of criteria for post-fire restoration prioritization. This would situate your work more firmly within the existing literature and highlight how your criteria (repetitive burns, slope, protected areas) align with or differ from approaches in previous research.

Additionally, as a suggestion for future work, it would be valuable to mention the potential for validation of your framework, for example by comparing the identified priority areas with outcomes of existing restoration actions in Greece or in other Mediterranean fire events. This would enhance the practical relevance and robustness of the proposed methodology.

Reviewer #2: The manuscript addresses an important topic and presents valuable data on prioritizing areas for post-fire restoration in Greece using a mixed-methods spatial approach. However, several aspects require substantive revision before the manuscript is suitable for publication.

Terminology consistency: The manuscript currently alternates between “wildfire,” “fire,” and “burns.” It would improve clarity and readability to use a consistent term throughout, e.g., “wildfire” for natural fires, and avoid switching to “burns.”

Abstract

I suggest not including the abbreviations GIS and RS in parentheses in the abstract. It is sufficient to use the full terms there and then introduce the abbreviations later in the main text, for example in the Materials and Methods section, as currently done.

Introduction

I noticed repeated citations of the same source – article no. 1 is cited in lines 41, 43, 58, and 61. It would be advisable to find additional sources for these statements, or to include more relevant references, especially considering the length and importance of these sections in the Introduction.

In lines 54–58, various damages caused by fires are mentioned, but it would be appropriate to also include damages to forests, since forests are frequently mentioned later in the text, as well as reforestation.

Use the term wildfire whenever referring to a forest fire. For example, in line 62 (both occurrences) and line 108. Of course, the terms fire severity and post-fire restoration are correctly used and do not require “wildfire.”

The sentence in lines 65–66 contains too many instances of “and.” I suggest rephrasing it, for example:

“The fires caused extensive damage to infrastructure, properties, and natural reserves, impacting biodiversity and local economies.”

In lines 68–69, it is stated that vegetation recovery generally occurs without intervention, with only a single citation (no. 11). I do not entirely agree with this, as replanting is frequently practiced (also mentioned in line 80 regarding reforestation in Greece). Either this statement should be supported by additional references, or the sentence should be reformulated.

From lines 88 to 99, I recommend standardizing the names of ministries and checking capitalization to ensure consistency.

In line 96, the term emergency stabilization should be clarified (stabilization of what?).

In line 100, “post-fire erosion risk” should be specified more clearly, e.g., soil erosion.

Material and methods

The manuscript uses the section title “Material and methods,” which is unusual. In last published PLOS ONE articles, the section is simply labelled “Methods,” although the journal’s submission guidelines recommend “Materials and Methods.” Consider adjusting the section title to align with typical journal usage.

Mixed methods section: The “Mixed methods” section reads more like part of the Introduction. In the Materials and Methods section, the focus should be on providing a detailed description of the procedures, data, analysis steps, selection of stakeholders, software used, and other methodological details. A graphical representation of the workflow would also be very helpful, as it would give the reader a clear overview of the process from stakeholder interviews to GIS analysis and the prioritization of areas. The paragraphs currently in lines 141–157 could be retained for context but should be moved to the Introduction, while the Materials and Methods should clearly outline the actual steps taken in the study.

Interviews section: I suggest reorganizing this section. First, it would be helpful to present the paragraph describing what was done in the study (currently lines 181–187), so that the reader clearly understands the procedures, topics, duration, and recording of the interviews. Only after that should the manuscript provide details on who was interviewed, including affiliations and the table currently in lines 161–178.

Rationale: If the reader first sees what was done (number of interviews, topics, duration, recording), they can better understand the context for the subsequent information on who was interviewed. This order allows readers to first grasp the methods and then see the additional details about the respondents, improving clarity and flow of the text.

Selecting recurrent criteria: This section clearly presents the criteria identified from the interviews, which is very helpful. I recommend adding a brief explanation of why only criteria mentioned by at least two respondents were included and how this threshold was determined.

Geospatial data: In this paragraph, it would improve clarity if the order of the data description matched the later table and detailed source section. For example, start with historical burned area data, then describe the DEM and slope data, and finally the protected areas. This ensures consistency and helps the reader follow the workflow from data selection to analysis. Additionally, it would be helpful to specify the time period covered by the historical burned area data, so that readers clearly understand the temporal scope of the dataset.

Abbreviation usage: Consider using the abbreviation GEE for Google Earth Engine in line 233 for consistency and brevity.

Results

Results clarity: The Results section clearly presents the outcomes of the analysis, but the description could be improved by consistently using terminology (e.g., “wildfire” instead of “fire” or “burned”) and by ensuring that figures and tables are referenced in a logical order matching the text.

It might improve readability if the presentation of results followed a consistent order: start with historical burned area data, then repetitive fires, slope analysis, and finally protected areas and overlay results. Additionally, providing percentages alongside absolute areas (as done in some cases) for all metrics would help readers quickly understand the relative scale of each criterion.

Discussion

Selected areas, Restoration measures, and Additional insights: Although these sections are labelled as Discussion, they mostly read as extended descriptions of results, case-specific examples, and practical recommendations, rather than analytical interpretation. They provide detailed accounts of selected areas, post-fire actions, community involvement, staffing, use of GIS, and funding challenges. In a typical Discussion, one would expect:

• comparison of findings with previous studies (e.g., “Our results are consistent with XY et al. (number)…”,

• interpretation of observed patterns and their ecological, social, or policy implications,

• acknowledgement of study limitations (currently addressed later in “Methodological remarks”), and

• broader implications for management, policy, or future research.

As currently written, the sections resemble Results or applied Recommendations. Reframing them to focus on analysis, interpretation, and synthesis would strengthen the manuscript and make it more aligned with standard scientific discussion structure.

Methodological remarks section: This is more like discussion, as it evaluates the approach, its strengths, limitations, and possibilities for extension. However, to constitute a discussion of manuscript, it would be important to compare the findings with previous studies, so that it is clear what is novel and what confirms or differs from existing knowledge.

I recommend adding statements such as:

• “Our findings are in line with XY et al. (number linked to references) who also highlighted the role of double-burned areas on steep slopes for restoration priorities.”

• “Unlike previous studies (AB et al., number linked to references), we integrated qualitative insights from stakeholders with geospatial analysis, providing a more nuanced perspective on post-fire management.”

• “The observed patterns of natural recovery correspond with findings reported by CD et al. (number linked to references) in Mediterranean ecosystems, emphasizing the importance of considering local socio-ecological constraints.”

This way, the section would not only evaluate the methodology and results but also situate the findings within the context of existing literature, which is exactly what a typical discussion should provide.

Conclusions:

The conclusion effectively summarizes the main findings of the study and emphasizes the practical applicability of the mixed-methods approach for prioritizing post-fire areas.

Overall, the manuscript has strong potential, but substantive revisions—especially in the Discussion and Introduction—are necessary to fully meet the standards of analytical interpretation and contextualization in the literature.

**Do you want your identity to be public for this peer review?** For information about this choice, including consent withdrawal, please see our Privacy Policy

Reviewer #1: No

Reviewer #2: No

---

## [Author Response · Author response to Decision Letter 2]

12 Dec 2025

See "Response to Reviewers" word document!

---

## [Editor Report · Decision Letter 2]

15 Dec 2025

Prioritizing Areas for Post-Fire Restoration in Greece Using Mixed-Methods Spatial Analysis

PONE-D-25-26522R2

Dear Dr. Palenova,

We’re pleased to inform you that your manuscript has been judged scientifically suitable for publication and will be formally accepted for publication once it meets all outstanding technical requirements.

Kind regards,

Kristofer Lasko, PhD

Academic Editor

PLOS One

Additional Editor Comments (optional):

The authors have revised the manuscript according to the minor revisions provided by the reviewers. However, the authors mentioned changing the title based on a suggestion from reviewer 1, but I do not see the title change. Please check and ensure the title is correct before publication.
---

## [Editor Report · Acceptance letter]

PONE-D-25-26522R2

PLOS One

Dear Dr. Palenova,

I'm pleased to inform you that your manuscript has been deemed suitable for publication in PLOS One. Congratulations! Your manuscript is now being handed over to our production team.

Kind regards,

on behalf of

Dr. Kristofer Lasko

Academic Editor

PLOS One